# Genome Mining Uncovers NRPS and PKS Clusters in *Rothia dentocariosa* with Inhibitory Activity against *Neisseria* Species

**DOI:** 10.3390/antibiotics12111592

**Published:** 2023-11-04

**Authors:** Elvis Achondou Akomoneh, Zina Gestels, Saïd Abdellati, Katleen Vereecken, Koen Bartholomeeusen, Dorien Van den Bossche, Chris Kenyon, Sheeba Santhini Manoharan-Basil

**Affiliations:** 1HIV/STI Unit, Department of Clinical Sciences, Institute of Tropical Medicine Antwerp, 2000 Antwerp, Belgium; eakomoneh@gmail.com (E.A.A.); zgestels@itg.be (Z.G.); sbasil@itg.be (S.S.M.-B.); 2Department of Microbiology and Parasitology, University of Bamenda, Bambili P.O. Box 39, Cameroon; 3Clinical Reference Laboratory, Department of Clinical Sciences, Institute of Tropical Medicine, 2000 Antwerp, Belgium; sabdellati@itg.be (S.A.); dvandenbossche@itg.be (D.V.d.B.); 4Virology Unit, Department of Biomedical Sciences, Institute of Tropical Medicine, 2000 Antwerp, Belgium; kvereecken@itg.be (K.V.); kbartholomeeusen@itg.be (K.B.); 5Department of Medicine, University of Cape Town, Cape Town 7700, South Africa

**Keywords:** NRPS, PKS, natural antimicrobials, *Rothia dentocariosa*, *Neisseria gonorrhoeae*

## Abstract

The growing global threat of antimicrobial resistance is reaching a crisis point as common bacterial infections, including those caused by pathogenic *Neisseria* species, are becoming increasingly untreatable. This is compelling the scientific community to search for new antimicrobial agents, taking advantage of computational mining and using whole genome sequences to discover natural products from the human microbiome with antibiotic effects. In this study, we investigated the crude extract from a *Rothia dentocariosa* strain with demonstrated antimicrobial activity against pathogenic *Neisseria* spp. by spot-on-lawn assay. The genomic DNA of the *R. dentocariosa* strain was sequenced, and bioinformatic evaluation was performed using antiSMASH and PRISM to search for biosynthetic gene clusters (BGCs). The crude extract with potential antimicrobial activity was run on Tricine-SDS-PAGE, and the putative peptides were characterised using liquid chromatography–tandem mass spectrometry (LC-MS). The crude extract inhibited the growth of the pathogenic *Neisseria* spp. Six BGCs were identified corresponding to non-ribosomal peptide synthases (NRPSs), polyketide synthases (PKSs), and ribosomally synthesised and post-translationally modified peptides. Three peptides were also identified corresponding to Actinorhodin polyketide putative beta-ketoacyl synthase 1. These findings serve as a useful reference to facilitate the research and development of NRPS and PKS as antimicrobial products against multidrug-resistant *N. gonorrhoeae.*

## 1. Introduction

The human oral microbiome harbours more than 700 bacteria species within different ecological niches and is considered as one of the most diverse in the human body [1]. As it is consistently in contact with the exterior environment, the oral cavity serves as one of the main portals of entry for pathogenic microorganisms, and therefore, a healthy oral microbiome could be an indispensable portion of the first line of defense against invading pathogens [1]. The ability of the human indigenous microbiota to resist colonisation and invasion by pathogenic bacteria is termed colonisation resistance [2]. Different mechanisms have been postulated to explain colonisation resistance, including the production of antagonistic compounds (e.g., bacteriocins and bacteriolytic enzymes) [3], competition for nutrients and adhesion sites [4], and the stimulation of the host immune response [5]. These activities create a microenvironment that prevents colonisation by and the dissemination of pathogens [6].

*Rothia* species are among the early colonisers of the oropharyngeal cavity, including the tonsils, tongue, and teeth of healthy humans [7,8] and are important members of the normal human oral microbiome [9,10,11]. *Rothia dentocariosa* is capable of fermenting a variety of carbohydrates and amino acids and is rarely implicated in human disease [12]. There is evidence that some *Rothia* species encode novel bioactive compounds in their biosynthetic gene clusters (BGCs) that could be used as a potential source of new antimicrobials [11]. Also, *Rothia* species have been identified with BGCs for archetype siderophore enterobactin, the strongest iron (III)-binding siderophore [12]. Biosynthetic gene clusters (BGCs) produce three major groups of metabolites: ribosomally synthesised and post-translationally modified peptides (RiPPs), non-ribosomal peptides (NRPs), and polyketides (PKS) [11,13]. RiPPs form a large heterogeneous group of peptides that are usually classified into post-translationally modified peptides and unmodified peptides [14]. NRPS-peptides are relatively small peptides containing 2–30 amino acids with hallmarks of non-proteinogenic amino acids, and their biosynthesis is ribosome independent [15,16]. Polyketides, on the other hand, are synthesised by polyketide synthases through repetitive condensations of acyl CoA esters to ward off competing microbes in response to nutrient limitation [17].

NRPS and PKS are the major enzymes of secondary metabolite synthesis that catalyse the synthesis of oligopeptides and the elongation of polyketides, respectively [18,19]. They can be extensively post-translationally modified, and these modifications lead to products with many features resembling the NRPS peptides [20]. Polyketides are classified into types I, II, and III [21] based on their biochemical mechanisms and enzyme architecture [22]. NRPS and PKS are responsible for the synthesis of a wide array of antimicrobial compounds, siderophores, and toxins, and they are valued clinically as antimicrobial, antifungal, antiparasitic, antitumor, and immunosuppressive agents [23]. Products and corresponding extracts from organisms with NRPS and PKS gene clusters have demonstrated interesting antimicrobial activities in vivo [6,11,13,24].

Previously, we isolated and identified *Rothia dentocariosa* from oropharyngeal swabs of men who have sex with men (MSM) with inhibitory activities against *Neisseria gonorrhoeae* and *N. meningitidis*, the two human pathogenic species of the genus *Neisseria* [25]. Considering the continuing emergence of antimicrobial resistance reported from both these pathogens, alternative treatments are urgently required [26,27]. In this study, we sought to expand our previous findings by sequencing the whole genome of the *R. dentocariosa* that we isolated and performed biosynthetic gene cluster (BGC) mining. The putative peptides were characterised using the nanoACQUITY LC system. We identified six BGCs that could possibly explain the inhibitory mechanisms employed by *R. dentocariosa* against the pathogenic *Neisseria* spp.

## 2. Methods

### 2.1. Samples Used in This Study and the Isolation and Identification of Rothia dentocariosa

Previously, we analysed oropharyngeal samples collected from men who have sex with men available from the Preventing Resistance to Gonorrhoeae (PReGo) study (ClinicalTrials.gov, NCT03881007) [28]. Briefly, bacterial isolates from 118 oropharyngeal swabs were cultured on BD^TM^ Columbia Agar with 5% sheep blood (BA) plates (Becton Dickinson, Heidelberg, Germany). Single colonies were obtained by plating a loopful of culture that was diluted (10^−6^) on BD^TM^ Columbia nalidixic acid agar (CNA) plates with 5% sheep blood (Becton Dickinson). This was followed by replica plating [29] of the colonies from each plate to gonococcal (GC) medium base (BD Difco™ Dehydrated Culture Media) agar plates (n = 3) that contained a lawn of *N. gonorrhoeae* (WHO P, reference isolate) or *N. meningitidis* (M00003, clinical isolate from the ITM collection). The replica plates were incubated overnight at 37 °C with 5% CO_2_. Inhibitory isolates were sub-cultured from the third set of replica plates, identified using matrix-assisted laser desorption/ionisation-time of flight (MALDI-TOF) Biotyper IVD (Bruker Daltonics, Bremen, Germany) (library updated to v.10.0.0.0_8326-9468 (IVD). Furthermore, growth inhibition of the inhibitory isolates was confirmed by agar overlay assay using the *N. gonorrhoeae* (WHO P) and *N. meningitidis* (M00003/1) isolates. This process led to the identification of a strain of *Rothia dentocariosa* that was inhibitory to the pathogenic *Neisseria* spp. [25].

### 2.2. Production of Crude Extracts from the Rothia dentocariosa Isolate

The protocol was adapted from Cheng et al. [30]. *R. dentocariosa* was grown in GC broth overnight for the production of crude extract. Briefly, 500 mL of GC semi-solid medium (GC broth supplemented with 6 g/L of agar) was prepared and allowed to cool for 24 h. This was stabbed with a sterile glass Pasteur pipet about 100 times, and 4 mL of the overnight culture was poured on the semisolid agar surface. The mixture was incubated at 37 °C with 5% CO_2_ in a CO_2_ incubator for 72 h. Following incubation, the semisolid agar was mashed with a spatula, dispensed into 50 mL falcon tubes, and stored overnight at −80 °C. The frozen mixture was then thawed for an hour in a water bath at 65 °C. The freeze/thaw cycle (1X) was used to disrupt the cell membranes of the *Rothia* bacteria. The thawed mixture was centrifuged at 4000× *g* for 140 min at 4 °C, and the supernatant was carefully collected and mixed with equal volumes of chloroform, vigorously shaken by hand for 20 min, and stored overnight at 4 °C. The upper layer was then removed, and the white interfacial layer was transferred into a beaker and kept in a fume hood (for approximately 3 days) for any residual chloroform to evaporate. The brownish extract-containing residue was dissolved in 7 mL of 35% acetonitrile-0.1% trifluoroacetic acid and centrifuged at 3200× *g* for 40 min at 4 °C to remove insoluble materials. The clear supernatant (henceforth referred to as crude extract) was passed through a 0.22 µm syringe filter, concentrated using a SpeedVac vacuum concentrator at room temperature, aliquoted, and stored at −20 °C for further analyses.

### 2.3. Detection of the Antimicrobial Activity of the Crude Extract

The antimicrobial activity of the crude extract was detected using the spot-on-lawn agar overlay assay protocol [25]. Briefly, 0.5 MacFarland of 24 h culture of target strains (*N. gonorrhoeae* or *N. meningitidis*) grown in GC broth was diluted in 4 mL of GC semi-solid medium pre-warmed to 55 °C to obtain a concentration of approximately 10^6^ CFU/mL. This was overlayed onto GC agar plates and allowed to solidify. Twenty microlitre aliquots of the crude extract were spotted on top of the agar overlay and allowed for complete adsorption. The plates were incubated at 37 °C with 5% CO_2_ in a CO_2_ incubator for 18 h and observed for the zone of inhibition. Twenty microlitres of 35% acetonitrile-0.1% trifluoroacetic was also spotted on an agar overlay as a negative control.

### 2.4. Separation of the Crude Extract by Tricine-SDS-PAGE Electrophoresis

The crude extract was run on sodium dodecyl sulphate–polyacrylamide gel electrophoresis (SDS-PAGE) to fractionate putative proteins [31]. Fifty microlitres of crude extract was mixed with 15 µL of sample loading dye and 3 µL of SDS-sample loading buffer (Sigma-Aldrich, Schnelldorf, Germany) and loaded into 4–20% Bis-Tris precast polyacrylamide gel (Bio-Rad, Hercules, CA, USA) alongside a low-range rainbow molecular weight marker (Spectra low molecular range marker, Thermo Scientific, Waltham, MA, USA) of sizes ranging from 3.4 kDa to 250 kDa. The setup was run at 140 V for 50 min in a mini vertical electrophoresis apparatus (Bio-Rad), after which the gel was stained with Coomassie brilliant blue G-250 (Thermo Scientific, Waltham, MA, USA). Following staining, the gel was washed (3×) with distilled water at room temperature to destain and then observed for visible protein bands. These bands were then sequenced for further analysis.

### 2.5. Whole-Genome Sequencing and Mining for Biosynthetic Gene Clusters (BGCs) in Rothia dentocariosa

The sequencing of *Rothia dentocariosa* was outsourced to Eurofins Genomics (Konstanz, Germany), where total DNA was isolated. Library preparation was carried out using a Stranded TruSeq DNA library preparation kit. Sequencing was performed on NextSeq6000, v2, 2 × 150 bp (Illumina Inc., San Diego, CA, USA), followed by analysis of the raw reads. For the WGS analysis, initial quality control (QC) of the raw reads was carried out using FASTQC [32]. The raw reads were trimmed using trimmomatic (v0.39) (Phred score ≥ 20 and length of the bases ≥32 bases) [33]. The processed raw reads were de novo assembled using Shovill (v1.0.4) [34], which uses SPAdes (v3.14.0) [35] using the following parameters: trim—depth 150—opts—isolate. The quality of the de novo assembled contigs was evaluated using Quast (v5.0.2) [36] and annotated using Prokka (v1.14.6) [37].

Further computational analysis was carried out as follows: briefly, the contigs were reordered using the *Rothia dentocariosa* ATCC 17931 genome sequence (GenBank accession CP002280.1), followed by joining the contigs using the ‘join sequences’ option implemented in CLC GenomicsWorkbench V20. The genome was run through the biosynthetic gene cluster (BGC) prediction program antiSMASH v 6.1.1 [38] (antismash --enable-genefunctions --enable-lanthipeptides --enable-lassopeptides --enable-nrps-pks --enable-sactipeptides --enable-t2pks --enable-thiopeptides --enable-tta --enable-html --fullhmmer --genefinding-tool prodigal -c 48 Rothia.gbk) and PRISM (https://prism.adapsyn.com/ (accessed on 4 July 2023)). The simplified molecular input line entry system (SMILES) and structures for the BGC products were predicted using PRISM [39]. The circular genome diagram was generated using CLC GenomicsWorkbench V20.

The raw reads generated were deposited at BioProject ID:PRJNA1013005.

### 2.6. Characterisation of Putative Peptides by Liquid Chromatography–Tandem Mass Spectrometry (LC-MS)

Bands on the SDS-PAGE gel were excised using a scalpel and collected on low-binding Eppendorf tubes (ThermoFisher scientific). The excised gel plugs were subjected to in-gel digestion following published protocols [40]. Briefly, the bands were washed with 100 µL water and shaken for 5 min. Water was then removed and replaced by 95% ACN to shrink the bands. The procedure was repeated several times to remove most of the staining solution. The bands were then swollen in 6.66 mM dithiothreitol dissolved in 50 mM ammonium bicarbonate (ABC, pH 7.8) and incubated at 56 °C for 45 min. They were then cooled to room temperature and washed with 95% ACN. The plugs were then swollen in 55 mM iodoacetamide, incubated in the dark for 30 min, and then washed with 95% ACN. Approximately 20 µL digestion buffer composed of 1 µg/µL trypsin suspended in 50 mM ABC was added and the tubes were incubated at 37 °C overnight. Digestion was stopped the following day by freezing the tubes. The extracts were then collected before and after the gel bands were shrunk in 95% ACN/H_2_O. These were dried under vacuum and resuspended in 20 µL 1% ACN/H_2_O. Approximately 2 µL was used to calculate the total peptide concentration using a NanoDrop^®^ ND-1000 UV-Vis Spectrophotometer (Thermo Scientific). Next, 0.5 µg of each sample (where possible; samples with lower peptide yields were all used) was loaded on a micropillar array (mPAC™) trapping column and injected on a 200 cm C18 mPAC™ column (Pharmafluidics, Zwijnaarde, Belgium) connected to a nanoACQUITY LC system (Waters). Separation was performed in reverse phase using a linear gradient of mobile phase B (0.1% formic acid in 98% acetonitrile) from 1% to 40% in 80 min, followed by a steep increase to 100% mobile phase B in 5 min. After 5 min at 100% mobile phase B, a steep decrease to 1% mobile phase B was achieved in 5 min, and 1% mobile phase B was maintained for 35 min. The flow rate was 750 nL per minute. The LC system was coupled to a Q-Exactive Plus orbitrap mass spectrometer (ThermoFisher Scientific) programmed to acquire in data-dependent mode. The survey scans were acquired in the orbitrap mass analyser operating at 70,000 (FWHM) resolving power at the mass range of 350–1850 m/z, with a target of 3E6 ions and 100 ms injection time. Precursors were selected “on the fly” for high-energy collision-induced dissociation (HCD) fragmentation with an isolation window of 1.6 amu and a normalised collision energy of 28%. A target of 1.7E3 ions and a maximum injection time of 80 ms were used for MS/MS. The method was set to analyse the top 20 most intense ions from the survey scan, and dynamic exclusion was enabled for 20 s.

Tandem mass spectra were processed using PEAKS Online X build 1.7.2022-08-03_160501 (Bioninformatics Solution, Inc., Waterloo, ON, Canada). Database searches were performed with the precursor and fragment tolerance set to 20 ppm and 0.05 Da, respectively, against an in-house database constructed from the protein translations of the whole-genome sequence of *R. dentocariosa*, as well as antiSMASH [38] identifications derived from the WGS. Trypsin was specified as the cleavage enzyme, allowing for a maximum of 2 miscleavages. Carbamidomethylation (C), and oxidation (M) and deamidation (NQ) were set as fixed and variable modifications, respectively. Peptide-to-spectrum matches (PSMs) were filtered at 1% FDR, with at least one peptide assigned per protein. Peptide de novo search was also performed, and the results were filtered at an average local confidence (ALC) ≥ 50%. PEAKS PTM and SPIDER, integrated with the PEAKS software, were also used to search for potential post-translational modifications and possible mutations using default search parameters.

## 3. Results

An overview of the study is provided in Figure 1.

### 3.1. Antibacterial Activity of the Crude Extract and Tricine-SDS-PAGE Analysis of Antibacterial Compounds

The crude extract and the concentrate that was obtained from *R. dentocariosa* showed inhibition against both the *N. gonorrhoeae* and *N. meningitidis* strains used in this study via the spot-on-lawn assay (Figure 2).

A prominent band with an apparent mass of around 80 kDa was identified for the crude extract (Figure 3).

### 3.2. The Rothia dentocariosa Genome Possesses Multiple Secondary Metabolite Biosynthetic Gene Clusters

The genomic DNA of *Rothia dentocariosa* was isolated and sequenced. We performed a bioinformatic evaluation of the *R. dentocariosa* genome using antiSMASH v.6.1.1 and PRISM to search for BGCs such as polyketide synthases (PKSs), non-ribosomal peptide synthases (NRPSs), and ribosomally synthesised and post-translationally modified peptides (RiPPs). Six biosynthetic gene clusters were identified and characterised (Figure 4 and Table 1). The BGCs corresponded to PKSs, NRPSs, and RiPPs and are as follows: one NRPS BGC was predicted to be related to the biosynthesis of enterobactin, EntF (Appendix A) and two genes belonging to one cluster were identified as type II PKS hybrid BGCs and related to the biosynthesis of Actinorhodin polyketide putative beta-ketoacyl synthase 1 and 2 (Appendix A). Additionally, two RiPPs, a lantipeptide-class-ii peptide (Appendix A) and a LAP (Linear azol(in)e-containing peptides) (Appendix A), along with Butyrolactone (Appendix A) were identified. Notably, out of the six BGCs, no homologous gene clusters could be identified for one cluster and hence represent an orphan BGCs (Figure 4).

### 3.3. Identification of Peptides

The crude extract with potential antimicrobial activity was run on Tricine-SDS-PAGE. The fractionated proteins were excised from the gel and the peptides were identified and characterised using the nanoACQUITY LC system for the presence of putative bacteriocins (peptides/proteins). As shown in Appendix A, three peptides, Ser-Gly-Phe-Ser-Glu-Glu-Glu-Ile-Ser-Thr-Leu-Asp-Lys, Glu-Ala-Glu-Met-Tyr-Gly-Ser-Ile-Thr-Gly-Tyr-Gly-Ala-Arg, and Ala-Tyr-Asp-Val-Ser-Ser-Leu-Lys (Table 1) were identified. The three peptides corresponded to Actinorhodin polyketide putative beta-ketoacyl synthase 1, identified by antiSMASH 6.0. No peptides were identified from any of the other five BGCs.

## 4. Discussion

As the search for novel approaches to contain multidrug-resistant bacteria continues, exploration of the mechanisms of colonisation resistance is increasingly gaining attention [41]. Previously, we identified a *Rothia dentocariosa* strain from oropharyngeal samples from men who have sex with men with inhibitory activity against pathogenic *Neisseria* spp. [25]. The crude extract derived from *R. dentocariosa* showed inhibitory activity against two pathogens, *Neisseria gonorrhoeae* and *Neisseria meningitidis*, as demonstrated in the spot-on-lawn assay (Figure 2).

In this study, we further characterised *R. dentocariosa* by sequencing the isolate and conducting in silico analyses of BGCs. More than 5000 BGCs have been identified in oral bacteria including *Rothia* species [11,42]. We identified six BGCs in this study corresponding to PKSs, NRPSs, and RiPPs whose products could be responsible for the inhibition of the pathogenic *Neisseria* spp. by *R. dentocariosa*. An in silico study by Isabela et al. 2022 [11], identified BGCs from 155 *Rothia* genomes predicted to produce antibiotic non-ribosomal peptides and other secondary metabolites. Similarly, using antiSMASH software, Uranga et al. 2020 [12] predicted that *R. dentocariosa* draft genomes (n = 12) harbored a butyrolactone BGC, a type 1 polyketide synthase (T1PKS) BGC, a lanthipeptide BGC, and a catechol siderophore-like (cat-sid) BGC. The same study identified an archetype siderophore enterobactin characterised from *R. mucilaginosa* ATCC 25296 that was shown to be involved in the reduced growth of some strains of *S. mutans*, oral *Streptococci* species, and oral *Actinomyces timomensis* [12]. In the present study, we identified enterobactin, EntF BGC (Appendix A), type II PKS hybrid BGCs, Actinorhodin polyketide putative beta-ketoacyl synthase 1 and 2 BGCs, two RiPPs, a lantipeptide-class-ii peptide (Appendix A), and a LAP (Linear azol(in)e-containing peptides) (Appendix A) BGCs, along with a Butyrolactone BGC (Appendix A). These peptides could explain some of *Rothia’s* competitive success in the oral cavity [11,12]. These results are therefore of great significance as PKS and NRPS gene clusters identified elsewhere are responsible for the synthesis of exciting novel antimicrobial compounds with a low risk of antimicrobial resistance, like valinomycin (from *R. nasisuis*) [11] and lugdinin (from *Staphylococcus lugdunensis*) [43].

Using the SDS-PAGE gel, a distinct band at approximately 80 kDa was identified that likely represents a bioactive molecule. By combining genome mining and liquid chromatography–tandem mass spectrometry (LC MS) analysis, we identified three peptides corresponding to Actinorhodin polyketide putative beta-ketoacyl synthase 1 and 2 in this band (Appendix A) which catalyse the biosynthesis of natural antimicrobials such as actinorhodin [44]. While these peptides are intriguing, they may represent only a fraction of the bioactive compounds present in the crude extract.

The existence of an orphan BGC within *Rothia dentocariosa’s* genome presents an opportunity for novel drug discovery. Similar orphan BGCs have been a focus of investigation in other studies [45]. Li et al. (2018) identified orphan BGCs in the *Streptomyces atratus* ZH16 genome, leading to the production of nocardamine-type (desferrioxamine) compounds via metabolic engineering, highlighting the importance of exploring uncharacterised gene clusters [46].

The caveats of the study include the following: we were unable to synthesise the peptides and prove that they had activity against the pathogenic *Neisseria* spp. We did however contact three companies/researchers linked to the compound in manuscripts and none could manufacture it for us. Also, we did not screen for antimicrobial activity against other bacteria genera in vitro and the bioactivity of the crude extract in vivo. Likewise, we did not characterise the secreted proteomes of this bacteria.

Nevertheless, this study provides novel evidence of the antibacterial activity of *Rothia dentocariosa’s* crude extract against *Neisseria* pathogens, offering a potential source of new antimicrobial agents. The genomic analysis sheds light on the biosynthetic potential of this commensal bacterium, and the discovery of an orphan BGC underscores the importance of further exploration. The identification of specific peptides raises exciting prospects for understanding the mechanisms of antimicrobial action within *Rothia dentocariosa* and highlights its potential contribution to combating antibiotic-resistant pathogens. Our findings highlight the potential for producing novel antimicrobial compounds from *Rothia dentocariosa*. Furthermore, we suspect that screening the nasal microbiome for bacteria that produce secondary metabolites will yield more peptides with anti-gonococcal and anti-meningococcal activity that could be developed as therapeutics.

## Figures and Tables

**Figure 1 antibiotics-12-01592-f001:**
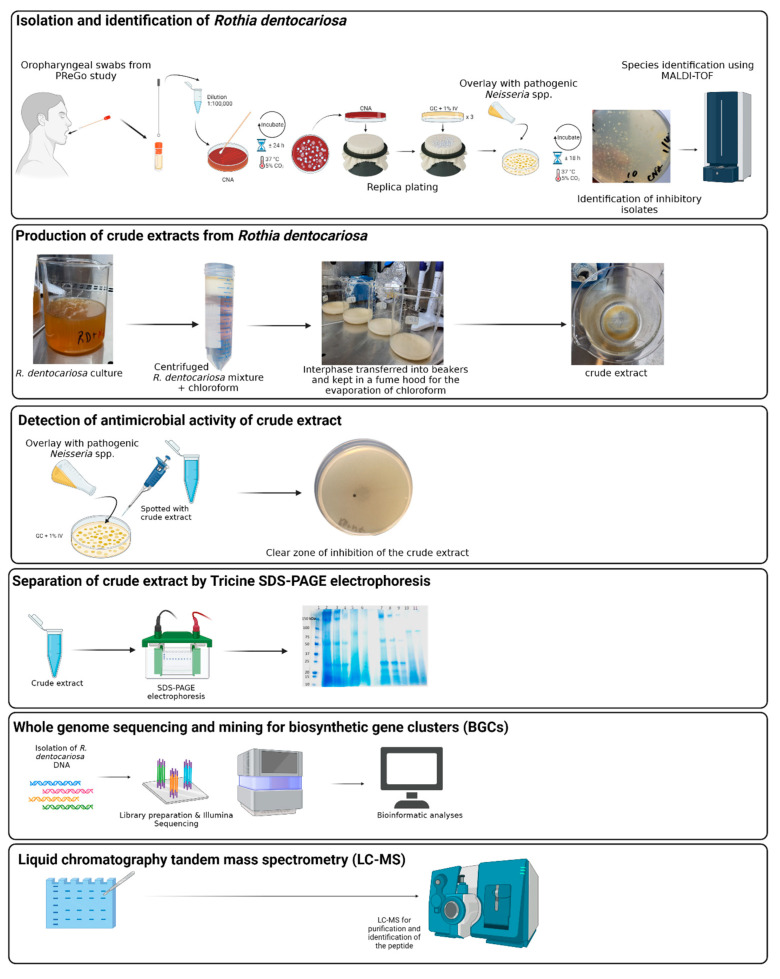
Overview of the study. The figure was generated using BioRender.

**Figure 2 antibiotics-12-01592-f002:**
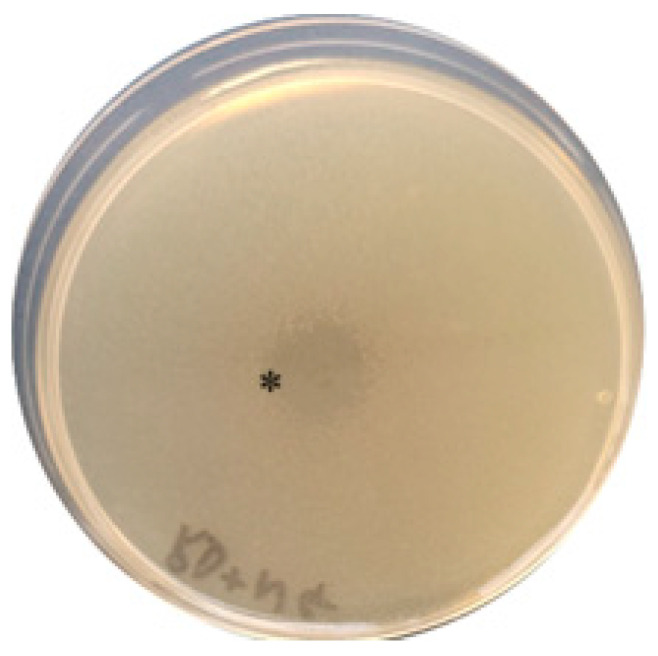
Spot-on-lawn assay showing the inhibitory effect (marked with *) of crude extract on *N. gonorrhoeae* growing on GC semi-solid medium.

**Figure 3 antibiotics-12-01592-f003:**
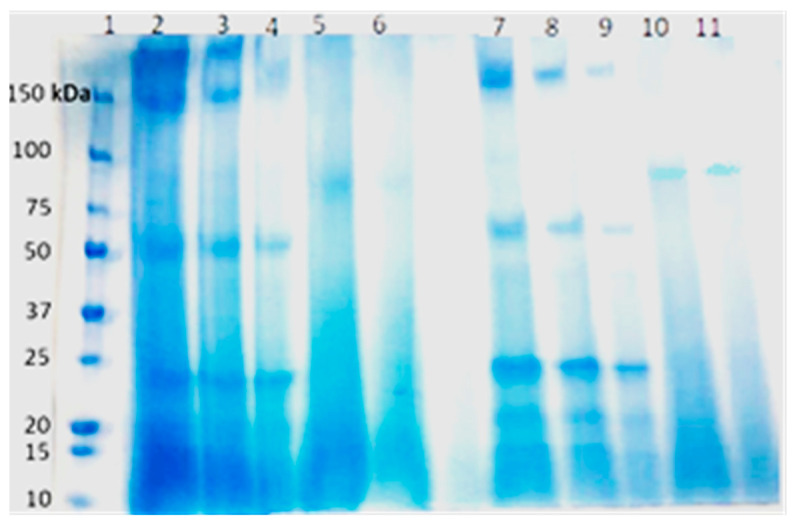
SDS–PAGE analysis of secretory products from *Streptococcus parasanguinis* and *Rothia dentocariosa***.** Marker (lane 1), *S. parasanguinis* (undiluted extract) (lanes 2–4), R. dentocariosa (undiluted extract) (lanes 5–6), *S. parasanguinis* (diluted 1:1) (lanes 7–9), and R. dentocariosa (diluted 1:1) (lanes 10–11).

**Figure 4 antibiotics-12-01592-f004:**
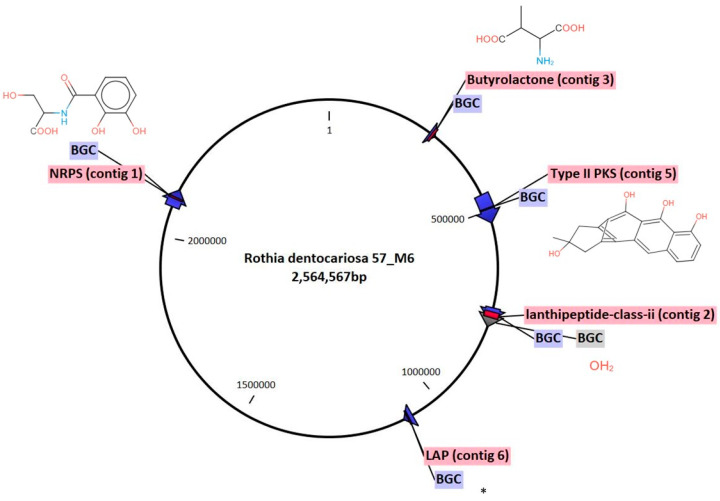
The *Rothia dentocariosa* (57_M6) genome possesses multiple biosynthetic gene clusters (BGCs). The BGCs were identified using antiSMASH v6.0 and the structures were predicted using PRISM. * No predicted structure available. The grey colored BGC depicts orphan the BGC.

**Table 1 antibiotics-12-01592-t001:** Putative secondary metabolites producing biosynthetic clusters of *Rothia dentocariosa* with homology to the most similar known biosynthetic clusters as predicted by antiSMASH 6.1.1.

Category	Type	Contig	Region	Start (bp)	End (bp)	Size (bp)	Core Biosynthetic Genes/Product	Locus Tag	Mass Spectrometry Identification (Separated Protein on Tricine SDS-PAGE Gel)
NRPS	NRPS	1	12.1	381,301	425,533	44,232	Enterobactin non-ribosomal peptide synthetase EntF	INOPIIDP_01825	Not detected
RiPP	lantipeptide-class-ii	2	4.1	130,371	178,145	47,774	Type 2 lanthipeptide synthetase LanM	INOPIIDP_00698	Not detected
							YcaO-like family protein	INOPIIDP_00709	Not detected
Other	Butyrolactone	3	2.1	259,764	275,626	15,862	AfsA-related	INOPIIDP_00236	Not detected
PKS	Hybrid (type II PKS)	5	3.1	128,334	200,795	72,461	Actinorhodin polyketide putative beta-ketoacyl synthase 2	INOPIIDP_00466	Not detected
							Actinorhodin polyketide putative beta-ketoacyl synthase 1	INOPIIDP_00467	Detected
RiPP	LAP	6	5.1	62,133	85,661	23,528	YcaO-like family protein	INOPIIDP_00975	Not detected
							SagB family peptide dehydrogenase	INOPIIDP_00976	Not detected

## Data Availability

The data supporting the findings of this study are retained at ITM and because of ethical and privacy concerns will not be made openly accessible. ITM adheres to the FAIR data principles (findable, accessible, interoperable, and reusable) and recognises that data should be “as open as possible and as closed as necessary”. Anonymised, individual participant data from the study as well as additional related documents, such as the study protocol, the annotated case report form, the data dictionary, and statistical analysis scripts, can be made available within 12 months of the publication of the study results and without an end date. Data will be retained at the ITM data repository and can be requested via email to ITM’s central point for research data access at ITMresearchdataaccess@itg.be. A governed data access mechanism applies including: (1) the completion of a data request form, (2) evaluation by a data access committee, (3) the signing of a data sharing agreement, and (4) the secure transfer of data.

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
