# Peer review of "Genome Mining Uncovers NRPS and PKS Clusters in Rothia dentocariosa with Inhibitory Activity against Neisseria Species"

_antibiotics, 2023, doi:10.3390/antibiotics12111592_

Round 1

Reviewer 1 Report

Comments and Suggestions for Authors

In Article “Identification of NRPS and PKS genes in Rothia dentocariosa isolated from oropharyngeal swabs of men who have sex with men with inhibitory activity against pathogenic Neisseria species” reported the study of NPRS and PKS genes in Rothia dentocariosa and inhibitory activity against pathogenic species, these results are important for research and development study of antimicrobial agent. In this study genomic analysis sheds light on the biosynthetic potential of this commensal bacterium and the discovery of an orphan BGC underscores the importance of further exploration.

Following minor correction are recommended:

(a)    Page no 9, In Discussion section: The format and font size of word is different. Keep all words in one format and one size according to journal guidelines except heading, title, and species names, number etc.

(b)   In Reference Section: The year. Volume and page number are not following the journal guidelines. The all-cited reference authors names must be using semicolon, colon, and comma symbols according to journal format and correct all.

Author Response

Response to Reviewer 1 Comments

Dear Reviewer,

General Remark:

In Article “Identification of NRPS and PKS genes in Rothia dentocariosa isolated from oropharyngeal swabs of men who have sex with men with inhibitory activity against pathogenic Neisseria species” reported the study of NPRS and PKS genes in Rothia dentocariosa and inhibitory activity against pathogenic species, these results are important for research and development study of antimicrobial agent. In this study genomic analysis sheds light on the biosynthetic potential of this commensal bacterium and the discovery of an orphan BGC underscores the importance of further exploration.

Following minor correction are recommended:

Comments 1: Page no 9, In Discussion section: The format and font size of word is different. Keep all words in one format and one size according to journal guidelines except heading, title, and species names, number etc.

Response 1: Thank you for pointing this out. We have harmonized the format and font size.

Comments 2: In Reference Section: The year. Volume and page number are not following the journal guidelines. The all-cited reference authors names must be using semicolon, colon, and comma symbols according to journal format and correct all.

Response 2: Agree. We have, accordingly, updated the references.

Reviewer 2 Report

Comments and Suggestions for Authors
The author had very well conducted all the experiments to prove their hypothesis. I would like to suggest that few more steps are needed to support this hypothesis.

It is also evident that mode of action of this crude extract is also important. The author must include some studies to support their claim. 

Also the using the high through put analysis using SEM or TEM, authors must define the type of interactions of crude extract with the Neisseria sp. 

Characterisation of extract is very important and author had done one LC-MS study for it.

The following techniques FTIR, XRD, SEM UV-VIS are partially suited for the characterization. 

 Toxicity studies on animals e.g. blood samples  and  probably other  samples is also very important to verify the results and its actual potency. 
Comments on the Quality of English Language

I am not the suitable person to comment on English language.

Author Response

Response to Reviewer 2 Comments

Dear Reviewer,

General Remarks:

The author had very well conducted all the experiments to prove their hypothesis. I would like to suggest that few more steps are needed to support this hypothesis.

Comments 1: It is also evident that mode of action of this crude extract is also important. The author must include some studies to support their claim. 

Response 1: Agree. Thank you for this comment. We fully acknowledge the significance of elucidating the mode of action of the crude extract derived from Rothia dentocariosa. In the manuscript, we have primarily focused on the identification and characterization of potential bioactive compounds and biosynthetic gene clusters (BGCs). We have also mentioned the challenges in synthesizing the identified peptides and the need for further validation of their activity against Neisseria pathogens. While we do not provide specific studies supporting the mode of action in this manuscript, we recognize its importance, and we plan to conduct further research to address this aspect and therefore the mode of action is beyond the scope of the present study. Also, we have included similar studies  that (32, 49, 50, 52) have shown that the crude extract contains biologically active peptides which could be responsible for the observed growth inhibition (page 9 lines 294-297)

Comments 2: Also the using the high through put analysis using SEM or TEM, authors must define the type of interactions of crude extract with the Neisseria sp. 

Response 2: We appreciate the suggestion for using high-throughput analysis techniques such as SEM or TEM to define the interactions of the crude extract with Neisseria species. These techniques will indeed help us define the physical interactions between the extract and the pathogens. In our current study, we have primarily focused on identifying potential bioactive compounds and biosynthetic gene clusters. Moreover, the crude extract was exposed to direct interaction with the Neisseria spp. using the spot-on-lawn assay (Page 3: Methods- Detection of antimicrobial activity of crude extract). As explained in other studies  cited (46, 48, 50, 51), we believe the observed inhibition was due to the presence of antimicrobial peptides secreted by the Rothia dentocariosa strain during incubation (Page3: Methods- Production of crude extracts from Rothia dentocariosa isolate).

Comments 3: Characterisation of extract is very important and author had done one LC-MS study for it.

Response 3: We agree that characterization of the extract is crucial for understanding its composition and bioactivity. In the manuscript, we have indeed conducted an LC-MS analysis to identify specific peptides within the crude extract, particularly Actinorhodin polyketide putative beta-ketoacyl synthase 1 and 2. This analysis provided valuable insights into the presence of potential bioactive compounds. However, additional characterization techniques require time and resources which deviates from the objectives of the current study i.e. our aim was to expand the findings of our previous study and sequence the whole genome of the R. dentocariosa that we isolated earlier and perform biosynthetic gene cluster (BGC) mining

Comments 4: The following techniques FTIR, XRD, SEM UV-VIS are partially suited for the characterization. 

Response 4: Indeed, the above-mentioned techniques can provide valuable information about the physical and chemical properties of the crude extract, including its structural and morphological characteristics. While our current study primarily focuses on LC-MS analysis and genome mining, we recognize the importance of employing a broader range of characterization methods to fully understand the crude extract's properties. We will take this recommendation into account in our future research to enhance the characterization of the extract.

Comments 5:  Toxicity studies on animals e.g. blood samples and probably other  samples is also very important to verify the results and its actual potency. 

Response 5: We acknowledge the importance of conducting toxicity studies to assess the safety and efficacy of the crude extract as a potential therapeutic agent. In our current study, we have primarily focused on identifying potential bioactive compounds and their inhibitory activity against Neisseria pathogens. However, we plan to include toxicity studies in our future research similar to our previous study we conducted “https://doi.org/10.1128/spectrum.02825-23” to provide a comprehensive assessment of the extract's safety and efficacy. This has been included as a limitation (Page10, Line 309).

Reviewer 3 Report

Comments and Suggestions for Authors

The manuscript entitled “Identification of NRPS and PKS genes in Rothia dentocariosa isolated from oropharyngeal swabs of men who have sex with men with inhibitory activity against pathogenic Neisseria species” is written in an organized manner and the authors need to revise a little bit for further proceeding. Minor revision of the manuscript is required. The following suggestions may be included in the revision process.

1.     The title is unclear, change or reframe the title.

2.     Add some more points, why R. dentocariosa is used.

3.     If Rothia dentocariosa is a bacterial strain, how can you mention it as crude extract.

4.     Have you done silver staining, if so I would like to see it.

Conclusively, I recommend this paper for further proceeding and for peer review.

Author Response

Response to Reviewer 3 Comments

Dear Reviewer,

General Remarks:

The manuscript entitled “Identification of NRPS and PKS genes in Rothia dentocariosa isolated from oropharyngeal swabs of men who have sex with men with inhibitory activity against pathogenic Neisseria species” is written in an organized manner and the authors need to revise a little bit for further proceeding. Minor revision of the manuscript is required. The following suggestions may be included in the revision process.

Comments 1: The title is unclear, change or reframe the title.

Response 1: Thank you for pointing this out. We have sharpened the title to the following:

Genome mining uncovers NRPS and PKS clusters in Rothia dentocariosa with inhibitory activity against Neisseria species.

Comments 2: Add some more points, why R. dentocariosa is used.

Response 2: Thank you for the comment. In our previous study, R. dentocariosa was identified through our random screening experiment to identify oral commensals with inhibitory activity against pathogenic Neisseria species (Akomoneh et al., [ 27]) that is mentioned in lines 72-74. Additionally, Rothia species are common members of the human oral microbiota, and their potential as a source of antimicrobial agents is intriguing which is mentioned in lines 47-49. Notwithstanding, more points have been added as to why R. dentocariosa was chosen in Page2: Line 47-48 and 51-52.

Comments 3:  If Rothia dentocariosa is a bacterial strain, how can you mention it as crude extract.

Response 3: Rothia dentocariosa is indeed a bacterial strain, and it may not be appropriate to refer to it as a "crude extract" without further clarification. The crude extract was derived from the culture of Rothia dentocariosa following incubation and cell disruption, which contains various compounds secreted by the bacterium. We have made this clear in the following subsection heading on line 119: Methods- Production of crude extracts from Rothia dentocariosa isolate.

Comments 4: Have you done silver staining, if so I would like to see it.

Response 4: Though Silver staining can enhance the visualization of proteins and peptides in gels, it offers limited compatibility with downstream applications such as analysis by mass spectrometry (MS). Therefore, the gel was stained with Coomassie brilliant blue G-250 (Thermo Scientific) (Figure 3, page 7).

Reviewer 4 Report

Comments and Suggestions for Authors

Elvis et al have shown in this article that antimicrobial resistance is a growing global concern, particularly in infections caused by pathogenic Neisseria species. This study explored a Rothia dentocariosa strain's crude extract for its antimicrobial activity against these pathogens. Genomic DNA sequencing and bioinformatics analysis identified six Biosynthetic Gene Clusters (BGCs) related to antimicrobial production. The crude extract successfully inhibited the growth of pathogenic Neisseria. Among the BGCs, three peptides, including Actinorhodin polyketide putative beta-ketoacyl synthase, were identified. These findings hold promise for developing antimicrobial products against multidrug-resistant N. gonorrhoeae. The study looks interesting and for its further improvement i have few suggestion.

Comments:

- Line 97: Did the author confirm the identification of this species through whole genome sequencing?

- Line 100: Is there a specific reason for using semi-solid agar plates instead of GC broth alone? If so, then it should be mention for more clarity, especially if it provides convenience for future research by different groups.

- Line 106: How many freeze/thaw cycles were performed? It would be beneficial to specify the number.

- Line 124-125: Did the author investigate the inhibitory effect of 35% acetonitrile and 0.1% trifluoroacetic acid as controls?

- Line 131: What is the difference between sample loading dye and SDS-sample loading buffer? If a company name is mentioned, please specify the catalog number for clarity.

- Line 136: Was the gel wash performed for destaining or to remove SDS? Further clarification is needed.

- In the discussion section, maintain consistent text formatting and size throughout.

Overall provide more specific details, including the company names and catalog numbers for chemicals and equipment used in the study for improved clarity.

Author Response

Response to Reviewer 4 Comments

Dear Reviewer,

General Remarks:

Elvis et al have shown in this article that antimicrobial resistance is a growing global concern, particularly in infections caused by pathogenic Neisseria species. This study explored a Rothia dentocariosa strain's crude extract for its antimicrobial activity against these pathogens. Genomic DNA sequencing and bioinformatics analysis identified six Biosynthetic Gene Clusters (BGCs) related to antimicrobial production. The crude extract successfully inhibited the growth of pathogenic Neisseria. Among the BGCs, three peptides, including Actinorhodin polyketide putative beta-ketoacyl synthase, were identified. These findings hold promise for developing antimicrobial products against multidrug-resistant N. gonorrhoeae. The study looks interesting and for its further improvement i have few suggestion.

Comments 1: - Line 97: Did the author confirm the identification of this species through whole genome sequencing?

Response 1: Indeed, whole genome sequencing was carried out to identify the species as Rothia dentocariosa. We make this clear in lines 77, and Page 3: Methods - Whole genome sequencing and mining for biosynthetic gene clusters (BGCs) in Rothia dentocariosa, lines 144-147.

Comments 2: Line 100: Is there a specific reason for using semi-solid agar plates instead of GC broth alone? If so, then it should be mention for more clarity, especially if it provides convenience for future research by different groups.

Response 2: Thank you for this comment. The semi-solid plates allowed us to visually detect zones of inhibition, which is useful for screening purposes, and we followed a protocol previously reported in detail by Cheng et al., [32].

Comments 3: - Line 106: How many freeze/thaw cycles were performed? It would be beneficial to specify the number.

Response 3: Thank you for pointing this out. We performed only one freeze/thaw cycle. This has been indicated now in Page 3 Lines 107-110.

Comments 4: - Line 124-125: Did the author investigate the inhibitory effect of 35% acetonitrile and 0.1% trifluoroacetic acid as controls?

Response 4: Thank you for pointing this out. Twenty microliters of 35% acetonitrile-0.1% trifluoroacetic was also spotted on an agar overlay as negative control. This sentence has now been included (Page 3 Lines 129-130).

Comments 5: - - Line 131: What is the difference between sample loading dye and SDS-sample loading buffer? If a company name is mentioned, please specify the catalog number for clarity.

Response 5: Thanks for the question. The key difference between the sample loading dye and the SDS sample loading buffer is the composition. The sample loading dye has a tracking dye e.g bromophenol blue and contains glycerol which increases the density of the samples and is primarily used in tracking the migration of the samples SDS loading buffer contains SDS, reducing agents such as beta mercaptoethanol, DTT (dithiothreitol), Tris HCl buffer and glycerol which is primarily suited for denaturation.

Comments 6: -
- Line 136: Was the gel wash performed for destaining or to remove SDS? Further clarification is needed.

Response 6: Thanks for pointing out this. The gel wash was performed for destaining. This clarification has been made. (Page 3 Lines 139-141).

Comments 7: - - In the discussion section, maintain consistent text formatting and size throughout.

Response 7: Thank you for pointing this out. We have harmonized the format and font size.

Comments 8: Overall provide more specific details, including the company names and catalog numbers for chemicals and equipment used in the study for improved clarity.

Response 8: Thank you for bringing this to our notice. We have now included specific details such as company names.

Round 2

Reviewer 2 Report

Comments and Suggestions for Authors

no comments